# Sampling Boltzmann distributions via normalizing flow approximation of transport maps

## Abstract

In a celebrated paper, Noé, Olsson, Köhler and Wu (Noé et al. (2019)) introduced an efficient method for sampling high-dimensional Boltzmann distributions arising in molecular dynamics via normalizing flow approximation of transport maps. Here, we place this approach on a firm mathematical foundation. We prove the existence of a normalizing flow between the reference measure and the true Boltzmann distribution up to an arbitrarily small error in the Wasserstein distance. This result covers general Boltzmann distributions from molecular dynamics, which have low regularity due to the presence of singular interatomic Coulomb and Lennard-Jones interactions. The proof is based on regularization of the interactions, a rigorous construction of the Moser transport map for Lipschitz endpoint densities, and approximation theorems for neural networks in Sobolev spaces.

Numerical simulations for a simple model system and for the alanine dipeptide molecule confirm that the true and generated distributions are close in the Wasserstein distance. Our simulations also reveal shortcomings of current RealNVP training protocols, caused by ignoring the singularities: large Lipschitz constants of the learned flows (as expected from our theoretical results; we prove that exact transport maps for molecular Boltzmann distributions are non-Lipschitz), inaccurate sampling near the singularities, and inaccurate estimation of observables like energy which are sensitive to the density near the singularities. Remedies are suggested which show promising performance in our numerical experiments.

## 1 Introduction

Our goal in this paper is to put the recent method of Noé et al. (2019) for sampling from high-dimensional Boltzmann distributions on a firm mathematical footing.

Sampling from Boltzmann distributions is of great interest in computational physics, chemistry, and biochemistry, because the conformations of a molecule are described by the dominant regions of the associated Boltzmann distribution. Sampling from this distribution is challenging for two reasons: first, the configuration space on which the distribution lives is high-dimensional, of dimension $3N$, where $N$ is the number of atoms in the molecule. Second, trajectory methods, i.e. using long-time samples of solutions of the underlying Langevin stochastic differential equation, while in principle correct, are expensive; this is because trajectories typically get trapped for a long time in metastable regions separated by energetic or entropic barriers, leading to very long simulation times for sufficient exploration of the configuration space.

Traditional approaches to this sampling problem include Markov state models Chodera et al. (2007), metadynamics Laio & Parrinello (2002), and umbrella sampling Torrie & Valleau (1977). While useful, these methods often either require careful hand-tuning of collective variables or suffer from high computational cost in high dimensions.

A novel approach based on ideas from machine learning was introduced in a pioneering paper by Noé et al. (2019), who proposed *Boltzmann generators*—invertible neural networks (normalizing flows) trained to map an easy-to-sample reference distribution (or prior) on the high-dimensional configuration space, for instance a Gaussian, to the target Boltzmann distribution. Note that, unlike in standard approaches in machine learning

for approximating high-dimensional distributions, *the domain of the prior is not dimension-reduced.* Once the flow is trained, a single forward pass through the flow of statistically independent samples drawn from the prior yields statistically independent equilibrium samples. This bypasses long-time stochastic trajectories and enables efficient exploration of metastable states. This approach builds on the rapid development of normalizing flows Dinh et al. (2017); Kingma & Dhariwal (2018); Papamakarios et al. (2021) and deep generative modeling for physical systems Hernandez et al. (2019); Wu et al. (2018).

Despite impressive empirical successes, a mathematical foundation of flow-based Boltzmann sampling is hitherto missing. Several open questions present themselves:

- **Dealing with singularities:** Physical Boltzmann distributions vanish on the collision set where two atoms collide, due to the singularities of the underlying Coulomb and Lennard-Jones potentials. Such zero regions, in turn, prevent the existence of smooth transport maps (for a proof of this interesting phenomenon see section 3). Thus it is not even clear whether any transport map exists between the Gaussian reference and the Boltzmann distribution which belongs to the regularity classes covered by expressivity results for neural networks.

- **Indeterminacy:** Even if singularities are absent or have been removed by some controlled approximation, the transport condition is highly degenerate. Which of the infinite-dimensionally many transport maps should be targeted for theoretical approximation results (and practical training)?

- **Approximability by normalizing flows:** Because of the need to efficiently evaluate the Jacobian of the transport map, a suitable class of neural networks is given by normalizing flow architectures such as RealNVP Dinh et al. (2017). Do these architectures possess sufficient expressive power to approximate the required transport maps between low-regularity densities well enough to produce correct samples?

The first problem is overcome by a simple regularization of the underlying molecular potential in high-energy regions, which entails a controlled $L^1$ error on the Boltzmann distribution. See section 3.

Regarding the second problem, a natural idea would be to use the Brenier map supplied by optimal transport theory Brenier (1991); Villani (2003); Friesecke (2024). For image generation, methods based on the Brenier maps from Gaussian noise to subsets of training images have proven useful Lipman et al. (2023); Liu et al. (2023); Chemseddine et al. (2025); Tong et al. (2023); Pooladian et al. (2023). Neural network based solvers for more general optimal transport problems have been introduced in Korotin et al. (2021); Uscidda & Cuturi (2023), based on input convex eural networks (ICNNs) respectively multi-layer perceptrons (MLPs). But the Brenier map, and the underlying flow, are not easy to compute between continuous distributions in high dimensions, and known regularity results for the map Caffarelli (1991); De Philippis & Figalli (2011) require strong assumptions on the densities. We therefore focus on a related but easier-to-compute and more regular transport map between continuous distributions, namely the Dacorogna-Moser map, and its underlying flow. We generalize its classical construction Moser (1965); Dacorogna & Moser (1990) from smooth densities to low-regularity densities (see Theorem 2 a)). We remark that for a uniform reference density, the governing problem correponds to a linearization of the optimal transport problem Friesecke (2024). The practical question of how to efficiently bias the learning of the flow towards the Dacorogna-Moser flow lies beyond the scope of this paper.

Finally, the third problem, rigorous approximability, is addressed in Theorem 2 c). We prove that the Moser map between Lipschitz continuous, everywhere positive densities can be approximated by normalizing flow architechtures such as RealNVP in the Sobolev space $W^{1,p}$.

Combining these results yields existence of a normalizing flow between the reference measure and the true Boltzmann distribution of a molecular system up to an arbitrarily small error in the Wasserstein distance (see Theorem 4), thus providing a theoretical guarantee that the Boltzmann generator method – under the idealized assumption that such a flow can be found by the training – gives correct samples.

Our theoretical results lead to the following practical question:

- **Practical effect of the singularities:** Current training protocols for RealNVP flows ignore the singularities. How does this affect the learned distributions in practice? Can the sampling performance be improved by taking the singularities into account during training?

This question is investigated in Section 6. We present numerical simulations for a simple synthetic system and for the alanine dipeptide molecule (22 atoms, 66-dimensional configuration space). Our results confirm that current training protocols give good overall agreement of the learned flows with the correct Boltzmann distribution in the Wasserstein distance. However, our results also reveal shortcomings caused by ignoring the singularities: learned flows exhibit large Lipschitz constants (as expected from Proposition 1), and produce inaccurate samples near the singularities. The latter leads, for alanine dipeptide, to a completely wrong predicted energy (off by a factor $2.5\times10^3$, see Table 2 in section 6.3); note that the energy, unlike the Wasserstein distance, is sensitive to the density in the small region near the singularities. Two remedies are proposed and tested: slightly regularize the singular interactions before training (in line with our theoretical results); or add a penalty to the loss function which teaches the normalizing flow to stay away from the singularities (easier to implement in practice as the original MD trajectories based on singular interactions can be used for training).

The question of how to achieve an optimally accurate and robust flow in practice, e.g. by biasing towards the Dacorogna-Moser flow associated with slightly regularized interactions, lies beyond the scope of this paper.

## 2 Molecular dynamics at thermal equilibrium

In equilibrium molecular dynamics, the state of a many-body system can be described by the Boltzmann probability distribution

$$d\rho(x) = \frac{1}{Z}\exp\left(-\frac{U(x)}{\kappa_B T}\right)dx, \tag{1}$$

where $x = (x_1, ..., x_N)$ with $x_i \in \mathbb{R}^3$ denoting the position of the $i$-th atom, $\kappa_B$ is a physical constant, $T$ is the temperature, and $U$ is the potential energy function of the system. One assumes that the position vector $x$ of all atoms is confined to some bounded domain $D \subset \mathbb{R}^{3N}$, and $Z$ is a normalization constant so that $\int_D \rho = 1$.

The potential energy function is typically taken to be of the form

$$\begin{aligned}
U(x) = &\sum_{\ell_{ij} \text{ bond lengths}} b_{ij}(\ell_{ij} - \overline{\ell_{ij}})^2 + \sum_{\alpha_{ijk} \text{ bond angles}} a_{ijk}(\alpha_{ijk} - \overline{\alpha_{ijk}})^2 \\
&+ \sum_{\phi_{ijkl} \text{ torsion angles}} \sum_{n=1}^{n_{max}} \kappa_{ijkl,n}\left[1 + \cos(n\phi_{ijkl} - \overline{\phi_{ijkl,n}})\right] \\
&+ \sum_{i,j} \frac{q_i q_j}{|x_i - x_j|} + \sum_{i,j}\left(\frac{A_{ij}}{|x_i - x_j|^{12}} - \frac{B_{ij}}{|x_i - x_j|^6}\right),
\end{aligned} \tag{2}$$

where the first three terms are summed, respectively, over the bond lengths, bond angles, and torsion angles in the bond graph of the molecular system and the last two terms are summed over all atoms in the system. In the above $q_i$, $A_{ij}$, $B_{ij}$, $a_{ijk}$, $b_{ij}$, $\kappa_{\phi_{ijkl,n}}$, $\overline{\ell_{ij}}$, $\overline{\alpha_{ijk}}$, and $\overline{\phi_{ijkl,n}}$ are constants specific to the molecular system. The constants $b_{ij}$ and $a_{ijk}$ are positive, $\kappa_{ijkl,n}$, $A_{ij}$ and $B_{ij}$ are nonnegative, and $A_{ij}$ must be positive when $q_i q_j < 0$ or $B_{ij} > 0$, and $x_i \in \mathbb{R}^3$ is the position of the $i$-th atom in the system. The last two terms in the energy functional represent electrostatic and Lennard-Jones interactions.

The energy equation 2 has various important qualitative properties:

$$\begin{aligned}
&U : \mathbb{R}^{3N} \to \mathbb{R} \cup \{+\infty\} \text{ is } C^1 \text{ on } \{x : x_i - x_j \neq 0 \,\forall i \neq j\}, \\
&U(x) \geq U_0 \text{ for some constant } U_0, \\
&U(x) \to \infty \text{ as } \min_{i \neq j}|x_i - x_j| \to 0.
\end{aligned} \tag{3}$$

The last property in equation 3 means that the potential energy tends to infinity when two atoms approach each other. As a consequence, the Boltzmann distribution equation 1 vanishes on the collision set $\{x_i = x_j$ for some $i \neq j\}$.

## 3 Transport maps and their lack of regularity

The Boltzmann generator approach for sampling from Boltzmann distributions starts from a smooth, positive, easy-to-sample reference distribution, or prior, and seeks to compute an invertible transport map between the reference distribution and the Boltzmann distribution equation 1. We begin by recalling basic mathematical definitions and facts. We then show that in our present context, transport maps must necessarily be quite irregular.

Assume that we are given two probability measures $\mu$ and $\nu$ on $\mathbb{R}^d$. (In the molecular dynamics case, $d = 3N$.) A *transport map* between $\mu$ and $\nu$ is a measurable map $T : \mathbb{R}^d \to \mathbb{R}^d$ satisfying

$$T_\sharp \mu = \nu. \tag{4}$$

The pushforward $T_\sharp \mu$ is defined such that for any measurable set $A \subset \mathbb{R}^d$, $(T_\sharp \mu)(A) = \mu(T^{-1}(A))$. Equivalently, for any measurable function $f$, we have

$$\int_{\mathbb{R}^d} f(y) \, d(T_\sharp \mu)(y) = \int_{\mathbb{R}^d} f(T(x)) \, d\mu(x).$$

This means that if a random variable $X \sim \mu$ then $T(X) \sim \nu$.

Assume now that $\mu$ and $\nu$ possess densities $\rho_0$ and $\rho_1$. In this case we write condition equation 4 as

$$T_\sharp \rho_0 = \rho_1. \tag{5}$$

If the map $T$ is invertible and its inverse is differentiable, by the change-of-variables formula eq. equation 4 is equivalent to

$$\rho_1(z) = \rho_0(T^{-1}(z)) \left| \det J_{T^{-1}}(z) \right|. \tag{6}$$

where $J_{T^{-1}}$ is the Jacobian matrix of $T$.

Invertible transport maps exist (in the low-regularity class of measurable mappings); a basic example is the Brenier map Brenier (1991); Villani (2003); Friesecke (2024) from optimal transport theory which exists provided the densities have finite second moments or compact support. But this map is not easy to compute in high dimension. Moreover known regularity results Caffarelli (1991); De Philippis & Figalli (2011) require the target density to be bounded away from zero, which is violated by Boltzmann distributions for molecules, due to equation 3.

Another, simpler and smoother, example of a transport map is the Dacorogna-Moser map, described in the next section; but its construction requires, among other assumptions, strict positivity of the target density, and therefore fails for Boltzmann distributions for molecules, which vanish on the collision set, due to equation 3.

In fact we show in Proposition 1 below that there exists *no* Lipschitz continuous tranwsport map from any continuous positive reference density to the Boltzmann distribution.

## 4 Normalizing flow approximation of transport maps

Throughout this section, we work in a general dimension $d$, which in the molecular dynamics setting corresponds to $d = 3N$.

Further, we assume that the map has the form of an $\ell$-fold composition

$$T = G^{(\ell)} \circ \cdots \circ G^{(1)}. \tag{7}$$

This composition structure emerges naturally for (i) the Brenier transport map which can be constructed via the Benamou–Brenier dynamic formulation of optimal transport, (ii) the Moser transport map Moser (1965); Dacorogna & Moser (1990) which is based on the flow of a more explicit vector field; see section 4.1.

In practice, the underlying flow and its step maps $G^{(k)}$ in equation 7 are not available explicitly.

The normalizing flow approach approximates each $G^{(k)}$ by a trainable invertible neural network block $F_{\theta_k}^{(k)}$,

$$G^{(k)} \approx F_{\theta_k}^{(k)},$$

(which itself consists of multiple coupling layers and alternating input partitions). The overall approximated transport map is then given by

$$F_\theta = F_{\theta_\ell}^{(\ell)} \circ F_{\theta_{\ell-1}}^{(\ell-1)} \circ \cdots \circ F_{\theta_1}^{(1)},$$

forming a neural-network discretization of the continuous transport. The resulting density is

$$\rho_\theta = (F_\theta)_\sharp \rho_0,$$

which approximates the target density $\rho_1$. Each $F_{\theta_k}^{(k)}$ is designed to be bijective with analytically computable inverse and Jacobian determinant.

The RealNVP architecture Dinh et al. (2017) provides a widely used framework for constructing such invertible transformations. For a RealNVP flow with $\ell$ coupling layers, the transformation reads

$$z = F_\theta(x) = F_{\theta_\ell}^{(\ell)} \circ F_{\theta_{\ell-1}}^{(\ell-1)} \circ \cdots \circ F_{\theta_1}^{(1)}(x).$$

For each layer $k = 1, \ldots, \ell$, let $d_a = \lfloor d/2 \rfloor$ and $d_b = d - d_a$. Split the input $x^{(k-1)} \in \mathbb{R}^d$ into two parts: $x_a^{(k-1)} \in \mathbb{R}^{d_a}$ (the part that is kept unchanged) and $x_b^{(k-1)} \in \mathbb{R}^{d_b}$ (the part that is transformed), where the roles of $x_a$ and $x_b$ are alternated from layer to layer. The forward transformation of the $k$-th layer is

$$\begin{cases} z_a^{(k)} = x_a^{(k-1)}, \\ z_b^{(k)} = x_b^{(k-1)} \odot \exp\big(s_{\theta_k}^{(k)}(x_a^{(k-1)})\big) + t_{\theta_k}^{(k)}(x_a^{(k-1)}), \end{cases}$$

with $x^{(0)} = x$ and $z = z^{(\ell)}$. The inverse mapping is explicit:

$$\begin{cases} x_a^{(k-1)} = z_a^{(k)}, \\ x_b^{(k-1)} = \big(z_b^{(k)} - t_{\theta_k}^{(k)}(z_a^{(k)})\big) \odot \exp\big(-s_{\theta_k}^{(k)}(z_a^{(k)})\big). \end{cases}$$

The Jacobian matrix of each layer is block-triangular:

$$J_{F^{(k)}}(x^{(k-1)}) = \begin{pmatrix} I_{d_a} & 0 \\ \dfrac{\partial z_b^{(k)}}{\partial x_a^{(k-1)}} & \mathrm{diag}\big( \exp(s_{\theta_k}^{(k)}(x_a^{(k-1)})) \big) \end{pmatrix},$$

with determinant

$$\det J_{F^{(k)}}(x^{(k-1)}) = \exp\left( \sum_{j=1}^{d_b} s_{\theta_k,j}^{(k)}(x_a^{(k-1)}) \right).$$

For the complete flow, the Jacobian determinant factorizes as

$$\det J_{F_\theta}(x) = \prod_{k=1}^{\ell} \det J_{F^{(k)}}(x^{(k-1)}).$$

This enables efficient density evaluation via the change-of-variables formula

$$\log \rho_\theta(z) = \log \rho_0(F_\theta^{-1}(z)) - \log \big| \det J_{F_\theta}(F_\theta^{-1}(z)) \big|.$$

### 4.1 Composition structure of the Brenier and Moser maps

We now describe how the composition structure equation 7 arises naturally for standard transport maps.

We begin with the Brenier map. Its basic static construction is as the optimizer $T : \mathbb{R}^d \to \mathbb{R}^d$ of $\int |x - T(x)|^2 \rho_0(x)\,dx$ subject to the constraint $T_\sharp \rho_0 = \rho_1$. Alternatively, it is the time-1 flow map of the time-dependent vector field $(v_t)_{t \in [0,1]}$, $v_t : \mathbb{R}^d \to \mathbb{R}^d$ which minimizes the action

$$\int_0^1 |v_t(x)|^2 \rho_t(x)\,dx\,dt$$

over velocity fields and density fields $(\rho_t)_{t \in [0,1]}$ subject to the continuity equation

$$\frac{\partial \rho_t}{\partial t} + \nabla \cdot (\rho_t v_t) = 0 \tag{8}$$

and the endpoint conditions $\rho_t|_{t=0} = \rho_0$, $\rho_t|_{t=1} = \rho_1$. The optimal density field is the displacement interpolation $\rho_t = \Phi_{t\sharp}\rho_0$, where $\Phi_t$ is the flow map of the optimal vector field. The flow map is $\Phi_t(x) = (1-t)x + tT(x)$, that is to say it moves mass along linear trajectories from a point $x$ to its image $T(x)$ under the Brenier map. In particular, $T = \Phi_1$, that is, it is the time-one map of a continuous flow.

The Moser map has the advantage that it is more explicit, but the price to pay is that the densities $\rho_0$ and $\rho_1$ must be assumed to be sufficiently smooth and strictly positive on a bounded domain with smooth boundary. The construction goes as follows (see Moser (1965); Dacorogna & Moser (1990), and see Friesecke (2024) for a textbook account). Define the velocity field

$$v_t(x) = \frac{\nabla u(x)}{(1-t)\rho_0(x) + t\rho_1(x)}, \tag{9}$$

where $u$ solves the following Poisson equation with Neumann boundary conditions

$$-\Delta u = \rho_1 - \rho_0 \text{ in } \Omega, \quad \nabla u \cdot n = 0 \text{ on } \partial\Omega. \tag{10}$$

Here $n(x)$ is the outward unit normal to $\partial\Omega$ at $x$. This construction guarantees that the linear density interpolation $\rho_t = (1-t)\rho_0 + t\rho_1$ satisfies the continuity equation equation 8. The velocity field $v_t$ gives rise to a flow map $\Phi_t : \Omega \to \Omega$, by solving the ordinary differential equation

$$\frac{d}{dt}\Phi_t(x) = v_t(\Phi_t(x)), \qquad \Phi_0(x) = x. \tag{11}$$

The map $T = \Phi_1$ is then a smooth diffeomorphism satisfying $(\Phi_1)_\sharp \mu = \nu$.

Finally, for both the Brenier map and the Moser map, the composition structure equation 7 is obtained by discretizing the time interval $[0,1]$ into $\ell$ uniform subintervals, each of length $1/\ell$, and introducing the flow maps $\Phi_{s,t}$ by replacing the initial condition in equation 11 by $\Phi_{s,t}(x)|_{t=s} = x$. The flow map over the $k$-th timestep is then, letting $t_k = k/\ell$,

$$G^{(k)}(x) = \Phi_{t_{k-1},t_k}(x) \approx x + \frac{1}{\ell}v_{t_k}(x), \qquad k = 1, \ldots, \ell,$$

providing the decomposition

$$T = G^{(\ell)} \circ G^{(\ell-1)} \circ \cdots \circ G^{(1)}.$$

## 5 Theoretical results

In the following section, we first show that existence of smooth (and even Lipschitz) transport maps fails for Boltzmann distributions for molecules, due to the singularities of the underlying Coulomb and Lennard-Jones potentials. We then provide a rigorous theoretical foundation for the Boltzmann generator method Noé et al. (2019), by combining a suitable regularization of the interatomic interactions, a rigorous construction of Moser transport for positive but low-regularity endpoint densities, and known approximation results for RealNVP flows.

### 5.1 Nonexsistence of Lipschitz transport maps

Here we show that the singularities of the interatomic interactions in molecular dynamics necessarily leads to singularities of any (exact) transport map. This fact is a challenge for theory, and also for efficient learning of such maps, on which more in section 6.

**Proposition 1** (Nonexistence of Lipschitz transport maps). *Let $\Omega \subset \mathbb{R}^{3N}$ be any open bounded set, let $\rho_0$ be any reference probability density on $\Omega$ which is continuous and strictly positive, and let $\rho_1$ be a Boltzmann distribution equation 1 on an open bounded set $D \subset \mathbb{R}^{3N}$ with $U$ satisfying equation 3. Then there does not exist any Lipschitz map $T$ which transports $\rho_0$ to the Boltzmann distribution $\rho_1$.*

**Proof.** Let $x^0 = (x_1^0, x_2^0, ..., x_N^0) \in \mathbb{R}^{3N}$ be any point with $x_1^0 = x_2^0$. For $r > 0$ and $\varepsilon > 0$, define the following neighborhood of this point:

$$B'_\varepsilon \times B'_r \times (B_r)^{N-2} := \{x \in \mathbb{R}^{3N} \ : \ |\tfrac{x_1 - x_2}{\sqrt{2}}| \leq \varepsilon, \ |\tfrac{(x_1 - x_1^0) + (x_2 - x_2^0)}{\sqrt{2}}| \leq r, \ |x_i - x_i^0| \leq r \ \forall i = 3, ..., N\}.$$

Now assume for contradiction that there exists an invertible Lipschitz map $T \ : \ \Omega^N \to \mathbb{R}^{3N}$ such that $T_\sharp \rho_0 = \rho_1$. Introduce the quantities

$$f_{min} := \min_{T^{-1}(B'_\varepsilon \times B'_r \times (B_r)^{N-2})} \rho_0, \qquad \eta(\varepsilon) := \max_{B'_\varepsilon \times B'_r \times (B_r)^{N-2}} \rho_1.$$

Note that the minimum is attained because the set over which it is taken is closed (since $T$ is continuous) and bounded (since $\Omega$ is). The key point is that Lipschitz maps can inflate the Lebesgue measure of any measurable set $A \subset \mathbb{R}^{3N}$ only by a factor, $\mathcal{L}(T(A)) \leq (\text{Lip}\,T)^{3N} \mathcal{L}(A)$. We exploit this as follows:

$$
\begin{aligned}
f_{\min} \cdot \mathcal{L}\big(B'_\varepsilon \times B'_r \times (B_r)^{N-2}\big) &= f_{\min} \cdot \mathcal{L}\big(T(T^{-1}(B'_\varepsilon \times B'_r \times (B_r)^{N-2}))\big) \\
&\leq f_{\min} \cdot (\text{Lip}\,T)^{3N} \mathcal{L}\big(T^{-1}(B'_\varepsilon \times B'_r \times (B_r)^{N-2})\big) \\
&\leq (\text{Lip}\,T)^{3N} \mu\big(T^{-1}(B'_\varepsilon \times B'_r \times (B_r)^{N-2})\big) \\
&= (\text{Lip}\,T)^{3N} \nu\big(B'_\varepsilon \times B'_r \times (B_r)^{N-2}\big) \\
&\leq (\text{Lip}\,T)^{3N} \mathcal{L}\big(B'_\varepsilon \times B'_r \times (B_r)^{N-2}\big) \cdot \eta(\varepsilon).
\end{aligned}
$$

But $f_{min}$ is positive whereas $\eta(\varepsilon)$ tends to zero as $\varepsilon \to 0$, a contradiction. $\qquad \square$

### 5.2 Regularization

Because of the nonexistence of smooth transport maps revealed by Proposition 1, we begin by introducing a regularization of the potential such that the error in the generated samples can be made arbitrarily small, depending on the choice of the regularization parameter.

**Theorem 1** (Convergence of regularized Boltzmann distributions). *For an arbitrary number $N$ of atoms, consider the configuration space $\Omega$, where $\Omega \subset \mathbb{R}^{3N}$ is open and bounded. Let $\beta > 0$, and let $U : \Omega \to \mathbb{R} \cup \{+\infty\}$ be a potential satisfing equation 2.*

a) *There exists a family of regularized potentials $\{U_\varepsilon\}_{\varepsilon>0}$, $U_\varepsilon : \Omega \to \mathbb{R}$, satisfying:*

    (i) ***Regularity:*** *$U_\varepsilon$ is $C^1$ and Lipschitz.*

    (ii) ***Uniform lower bound:*** *There exists $M > 0$ such that $U_\varepsilon(x) \geq -M$ for all $x \in \Omega$ and all $\varepsilon > 0$.*

    (iii) ***Pointwise convergence:*** *For all $x \in \Omega$, $\lim_{\varepsilon \to 0} U_\varepsilon(x) = U(x)$.*

    (iv) ***No change except at high energy:*** *$U_\varepsilon(x) = U(x)$ for all $x$ with $U(x) \leq \frac{1}{\varepsilon}$.*

b) *Define the normalized Boltzmann densities*

$$\rho_\varepsilon(x) = \frac{F_\varepsilon(x)}{Z_\varepsilon}, \qquad \rho(x) = \frac{F(x)}{Z}, \tag{12}$$

*with $F_\epsilon(x) = e^{-\beta U_\epsilon(x)}$, $F(x) = e^{-\beta U(x)}$, and the partition functions*

$$Z_\varepsilon = \int_\Omega F_\varepsilon(x)\,dx, \qquad Z = \int_\Omega F(x)\,dx.$$

*Then $\rho_\varepsilon$ is $C^1$ and Lipschitz, and*

$$\lim_{\varepsilon \to 0} \|\rho_\varepsilon - \rho\|_{L^1(\Omega)} = 0.$$

A important feature of the above regularization is property (iv). Thus the potential, the molecular dynamics trajectories, and the Boltzmann distribution are unchanged except in regions of very high energy. At room temperature, trajectories will in practice not reach such energies and the above regularization is only needed for rigorous theory but not for practical learning of normalized flows.

**Proof.** For $\lambda > 0$, define the set $K_\lambda = \{x \in \Omega : U(x) > \lambda\}$. Choose a cutoff function $\zeta \in C^1(\mathbb{R}^{3N})$ with $0 \le \zeta \le 1$ which is 1 on $K_{2/\varepsilon}$ and 0 outside $K_{1/\varepsilon}$, and let $U_\varepsilon(x) = (1 - \zeta(x))U(x) + \zeta(x) \cdot \frac{2}{\varepsilon}$. Then (i) and (iv) are satisfied. Moreover $U_\varepsilon$ is everywhere finite, $C^1$, and globally Lipschitz, and so is $\rho_\varepsilon$. Since the $U_\varepsilon$ are uniformly bounded below and converge pointwise to $U$, the Boltzmann factors $F_\varepsilon = e^{-\beta U_\varepsilon}$ converge pointwise to $F = e^{-\beta U}$. The bound $F_\varepsilon \le e^{\beta M}$ allows application of the dominated convergence theorem, implying $Z_\varepsilon \to Z$ and $F_\varepsilon \to F$ in $L^1(\Omega)$. Hence $\rho_\varepsilon = F_\varepsilon/Z_\varepsilon \to \rho = F/Z$ in $L^1(\Omega)$, completing the argument. $\square$

### 5.3 Moser transport map: existence and approximation by RealNVP flow

We now prove that the Moser transport map exists between low-regularity endpoint densities, thereby extending the classical construction in a smooth setting Moser (1965); Dacorogna & Moser (1990). We then show Sobolev approximability of the map by invertible neural networks.

**Theorem 2** (Existence and RealNVP approximation of the Moser transport map)**.** *Let $\Omega \subset \mathbb{R}^d$ be a bounded $C^3$ domain. Let $\rho_0$ and $\rho_1$ be Lipschitz continuous probability densities on $\Omega$ which are strictly positive with $\inf_\Omega \rho_0 > 0$, $\inf_\Omega \rho_1 > 0$.*

a) *There exists a solution $u \in W^{2,2}(\Omega)$ to the Neumann problem equation 10, and this solution belongs to $W^{3,p}(\Omega)$ for all $p > 1$. The corresponding Moser vector field equation 9 is Lipschitz. The associated Moser transport map $T = \Phi_1$ defined by equation 11 is an invertible bilipschitz map and transports $\rho_0$ to $\rho_1$, that is to say it satisfies equation 4.*

*If in addition $\rho_0$ and $\rho_1$ are $C^1$, then:*

b) *The Moser transport map $T = \Phi_1$ is a $C^1$-diffeomorphism.*

c) *For every $p > d$ and every $\varepsilon > 0$, there exists an invertible RealNVP neural network $F_\theta : \Omega \to \Omega$ composed of a finite number of coupling layers such that*

$$\|F_\theta - T\|_{W^{1,p}(\Omega)} < \varepsilon.$$

**Proof.** We begin with a). Existence of a $W^{2,2}$ solution to the Neumann problem equation 10 is a standard fact of PDE theory, noting that $\rho_1 - \rho_0$ belongs to $L^2(\Omega)$ and integrates to 0. Higher $W^{3,p}$ regularity is proved in Grisvard (2011) Lemma 2.4.2.2 and Theorem 2.5.1.1, noting that $\rho_1 - \rho_0$ belongs to $W^{1,p}(\Omega)$. This higher regularity means $\nabla u \in W^{2,p}$, and the Sobolev embedding theorem for $p > d$ gives $\nabla u \in C^1(\overline{\Omega})$. In particular, $\nabla u$ is Lipschitz. Together with the assumptions on $\rho_0$ and $\rho_1$ this yields that the vector field $v_t$ defined in equation 9 is Lipschitz in $x$ and continuous in $t$. It is then a standard fact of ODE theory that the associated flow map $\Phi_t : \Omega \to \Omega$ exists and is an invertible bilipschitz map for any $t$. By construction $T$ transports $\rho_0$ to $\rho_1$. b) follows from the fact that under the additional assumption made, the vector field equation 9

is $C^1$. As regards c), RealNVP coupling flows are dense with respect to the $W^{1,p}$ topology in the class of $C^1$-diffeomorphisms on compact domains Ishikawa et al. (2023). For an earlier, related approximation result in Sobolev spaces (namely approximability of Lipschitz functions by ReLu networks in the $W^{s,p}$ topology for all $s < 1$) see Gühring et al. (2020). Thus $T$ can be approximated arbitrarily well in $W^{1,p}(\Omega)$ by a RealNVP map $F_\theta$. Finally, being a RealNVP map, $F_\theta$ is invertible. □

### 5.4 Convergence of push-forward density

Next we prove convergence of the push-forward density under the RealNVP flow, in the narrow sense. This relies on the $W^{1,p}$ convergence result from Theorem 2, the Sobolev embedding theorem, and simple estimates.

Recall that a sequence of probability distributions $\rho_j$ on any closed subset of $\mathbb{R}^d$ converges narrowly to a probability distribution $\rho$ on $\Omega$ if

$$\int_\Omega f(y)d\rho_j(y) \to \int_\Omega f(y)d\rho(y) \text{ for all } f \in C_b(\Omega), \tag{13}$$

where $C_b(\Omega)$ is the space of bounded continuous functions on $\Omega$.

**Theorem 3** (Convergence of push-forward densities)**.** *Let $\Omega$, $\rho_0$, $\rho_1$ be as in Theorem 2 b). Then there exists a sequence of invertible realNVP neural networks $(F_\theta)_j : \Omega \to \Omega$ such that*

$$(F_\theta)_{j\sharp}\rho_0 \to \rho_1 \text{ narrowly as } j \to \infty.$$

The theorem says that bounded continuous observables are approximated correctly by the RealNVP flow. Here $j$ indexes the sequence; as $j$ increases the networks $(F_\theta)_j$ approximate the target more closely, at the expense of containing more and more coupling layers.

We conjecture that such an approximation result is also possible in the strong $L^1$ sense. This would follow if the RealNVP map $F_\theta$ from Theorem 2 could be constructed in such a way that, in addition, its inverse $F_\theta^{-1}$ approximates $T^{-1}$ arbitrarily well in $W^{1,p}(\Omega)$. Such a result with $p = d$ would imply that the inverse Jacobian appearing in equation 6 is approximated in $L^1(\Omega)$.

**Proof.** Let $T : \Omega \to \Omega$ be the Moser map between $\rho_0$ and $\rho_1$ from Theorem 2 a), b). By Theorem 2 c) and the Sobolev embedding $W^{1,p}(\Omega) \hookrightarrow C^0(\overline{\Omega})$ for $p > d$, there exists a sequence of invertible RealNVP neural networks $(F_\theta)_j$ such that

$$\sup_{x \in \Omega} |(F_\theta)_j(x) - T(x)| \to 0 \ (j \to \infty). \tag{14}$$

For any given $f \in C_b(\Omega)$ we now calculate using the change-of-variables formula for the push-forward

$$\left| \int f \, d(F_\theta)_{j\sharp}\rho_0 - \int f \, d\rho_1 \right| = \left| \int f \, d(F_\theta)_{j\sharp}\rho_0 - \int f \, dT_\sharp\rho_0 \right| = \left| \int f((F_\theta)_j(x))d\rho_0(x) - \int f(T(x)) \, d\rho_0(x) \right|$$

$$\leq \sup_{x \in \Omega} \left| f((F_\theta)_j(x)) - f(T(x)) \right| \int 1 \, d\rho_0 \to 0 \ (j \to \infty),$$

where the convergence in the last line is due to the fact the first factor converges to zero thanks to equation 14 and the continuity of $f$, and the second factor is equal to 1 since $\rho_0$ is a probability measure. Since $f$ was arbitrary, this establishes the asserted narrow convergence equation 13. □

### 5.5 Main result

Finally we are in a position to state our main result.

**Theorem 4** (Theoretical justification of the Boltzmann generator method)**.** *For an arbitrary number $N$ of atoms, any bounded domain $\Omega \subset \mathbb{R}^{3N}$ of class $C^3$, any inverse temperature $\beta > 0$, any potential $U : \Omega \to \mathbb{R} \cup \{+\infty\}$ satisfing equation 2, any $C^1$ and Lipschitz reference probability density on $\Omega$ which is strictly*

*positive with $\inf_\Omega \rho_0 > 0$, and any $\epsilon > 0$, there exists an invertible RealNVP neural network $F_\theta : \Omega \to \Omega$ such that*

$$W_2(F_{\theta\sharp}\rho_0, \rho_1) < \varepsilon,$$

*where $\rho_1$ is the Boltzmann distribution equation 1 and $W_2$ denotes the Wasserstein-2 distance.*

**Proof.** This follows by combining Theorems 1, 2, 3 and the fact (see e.g. Villani (2003); Friesecke (2024)) that the Wasserstein-2 distance metrizes narrow convergence of probability measures on compact sets. $\square$

## 6 Numerical Experiments

In this section, we illustrate the Boltzmann generator method and the practical effect of the singular interactions for three model problems (1) overdamped Langevin dynamics for two particles in a four-dimensional double-well potential with Coulomb repulsion, (2) molecular dynamics for the alanine dipeptide molecule projected onto collective variables, (3) molecular dynamics for alanine dipeptide in full 66-dimensional coordinate space.

### 6.1 Two-particle system with a Coulomb singularity

We consider two particles in the plane with positions $(x_1, x_2)$ and $(y_1, y_2)$ and a (singular) Coulomb repulsion, and examine the impact on learned distributions of regularizing the interaction before training. The potential energy is

$$U(x) = (x_1^2 - 1)^2 + x_2^2 + (y_1^2 - 1)^2 + y_2^2 + \frac{\lambda}{r}, \qquad r = \sqrt{(x_1 - y_1)^2 + (x_2 - y_2)^2}, \tag{15}$$

with $\lambda = 0.7$. The double-well terms keep each particle near $x_1 = \pm 1$ and $y_1 = \pm 1$, while the $1/r$ term diverges when the two particles approach each other, exhibiting a singularity at $r = 0$ where the Boltzmann distribution $\sim e^{-\beta U}$ vanishes. Therefore by Proposition 1 there exists no Lipschitz transport map from the Gaussian reference to the Boltzmann distribution.

To illustrate the effect of the singularity on the training of the Boltzmann generator, we also consider the regularised potential obtained by replacing the singular Coulomb term with a soft-core form

$$U_\varepsilon(x) = (x_1^2 - 1)^2 + x_2^2 + (y_1^2 - 1)^2 + y_2^2 + \frac{\lambda}{\sqrt{r^2 + \varepsilon^2}}. \tag{16}$$

For any $\varepsilon > 0$, $U_\varepsilon$ is $C^1$, globally Lipschitz, and bounded, and it converges pointwise to the singular potential as $\varepsilon \to 0$, in the spirit of Theorem 1. The resulting Boltzmann distribution $\sim e^{-\beta U_\epsilon}$ is strictly positive, and by Theorem 2 there now exists a Lipschitz transport map from the Gaussian reference to the Boltzmann distribution.

a) **Original Boltzmann generator method.** First, we generated $12,000$ equilibrium samples at inverse temperature $\beta = 5$ from the singular reference distribution $\sim e^{-\beta U}$ (approximated with $\varepsilon = 10^{-8}$), using a simple MCMC sampler. We then trained a RealNVP flow directly on the singular reference. After training, we drew $10,000$ new samples from the flow and compared to the singular reference. The results are shown in Figure 1, left panel, and Table 1.

b) **Regularized Boltzmann generator method.** We picked three values $\varepsilon \in \{0.03, 0.1, 0.5\}$, and generated $10,000$ equilibrium samples at inverse temperature $\beta = 5$ from the regularized reference distribution $\sim e^{-\beta U_\varepsilon}$. For each, a RealNVP flow with 8 coupling layers and 256 hidden units (total 1.09M parameters) was trained on the regularised data for 400 epochs. After training, we drew $10,000$ new samples from each flow and compared them to both the regularized and the singular reference distribution. The results are shown in Figure 1, 2nd to fourth panel, and Table 1.

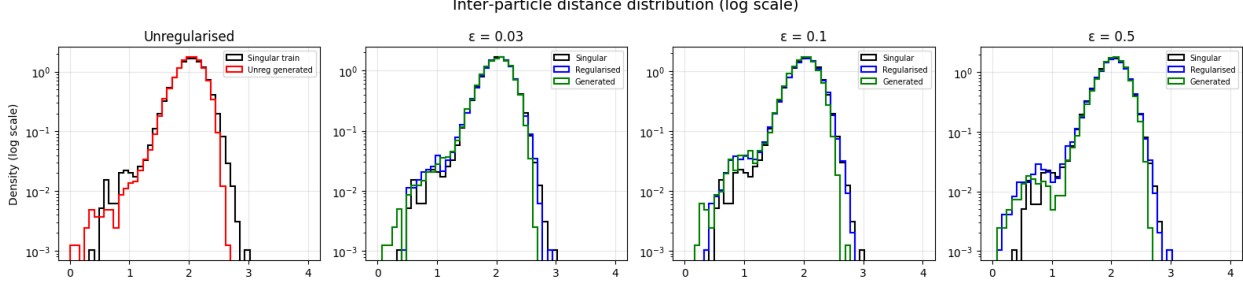

Figure 1: Inter-particle distance distributions. **Left panel:** singular reference training data (black), unregularised generated samples (red). **Remaining panels:** singular reference training data (black), soft-core regularised training data with $\varepsilon = 0.03, 0.1, 0.5$ (blue), generated samples (green).

Table 1: Results for the two-particle Coulomb system with soft-core regularisation ($\beta = 5$, $\lambda = 0.7$).

| Case | Min $r$ (real) | Mean $r$ (real) | Min $r$ (gen) | Mean $r$ (gen) | $W_2(sing, gen)$ | Lip |
|---|---|---|---|---|---|---|
| Unregularised | 0.401 | 2.005 | 0.077 | 1.998 | 0.017 | 87.6643 |
| $\varepsilon = 0.03$ | 0.475 | 1.996 | 0.148 | 1.986 | 0.023 | 59.1671 |
| $\varepsilon = 0.1$ | 0.379 | 1.991 | 0.180 | 1.969 | 0.047 | 51.1226 |
| $\varepsilon = 0.5$ | 0.084 | 1.998 | 0.155 | 2.005 | 0.016 | 25.2612 |

Table 1 and Figure 1 reveal the following effects. The flow trained on unregularized data is close in the Wasserstein-2 distance to the reference data in line with what is expected from our theoretical findings (Theorem 4). However the flow generates a minimum inter-particle distance of 0.077, while the training minimum is 0.401; it clearly fails to learn the repulsive barrier. With the soft-core regularisation at $\varepsilon = 0.1$, the generated minimum distance rises to 0.180, closer to the regularised training minimum of 0.379, indicating a better but still not fully satisfactory learning of the repulsive barrier, while the Wasserstein-2 distance to the singular reference data remains small (0.047). When $\varepsilon$ is taken too large, at 0.5, the repulsive barrier is no longer captured by the training data nor by the generated flow. In conclusion, a small amount of regularization yields some improvement of the method in the low-density region.

As $\varepsilon$ increases, the Lipschitz constant of the learned flow drops from 87.7 (unregularised) to 51.1 ($\varepsilon = 0.1$) and further to 25.3 ($\varepsilon = 0.5$). This numerically confirms our theoretical finding that an exact matching of the singular reference requires Lipschitz constant $+\infty$ (Proposition 1), whereas an exact matching of the distribution for a regularized potential is possible with finite Lipschitz constant (Theorem 2).

Figure 2 shows $x_1$ trajectories from a real Langevin simulation and a sequence generated by the normalizing flow. The generated sequence accurately reproduces the switching dynamics between the two wells, indicating that the RealNVP architechture is capable of capturing not just the equilibrium distribution but also the meastable dynamics.

In summary, we conclude that both the original and the regularized method yield good overall agreement between the true Boltzmann distribution and the generated samples. The regularized method shows some improvement in capturing the low-density region. Moreover the regularized method leads to a more regular flow, in the sense of a significantly lower Lipschitz constant.

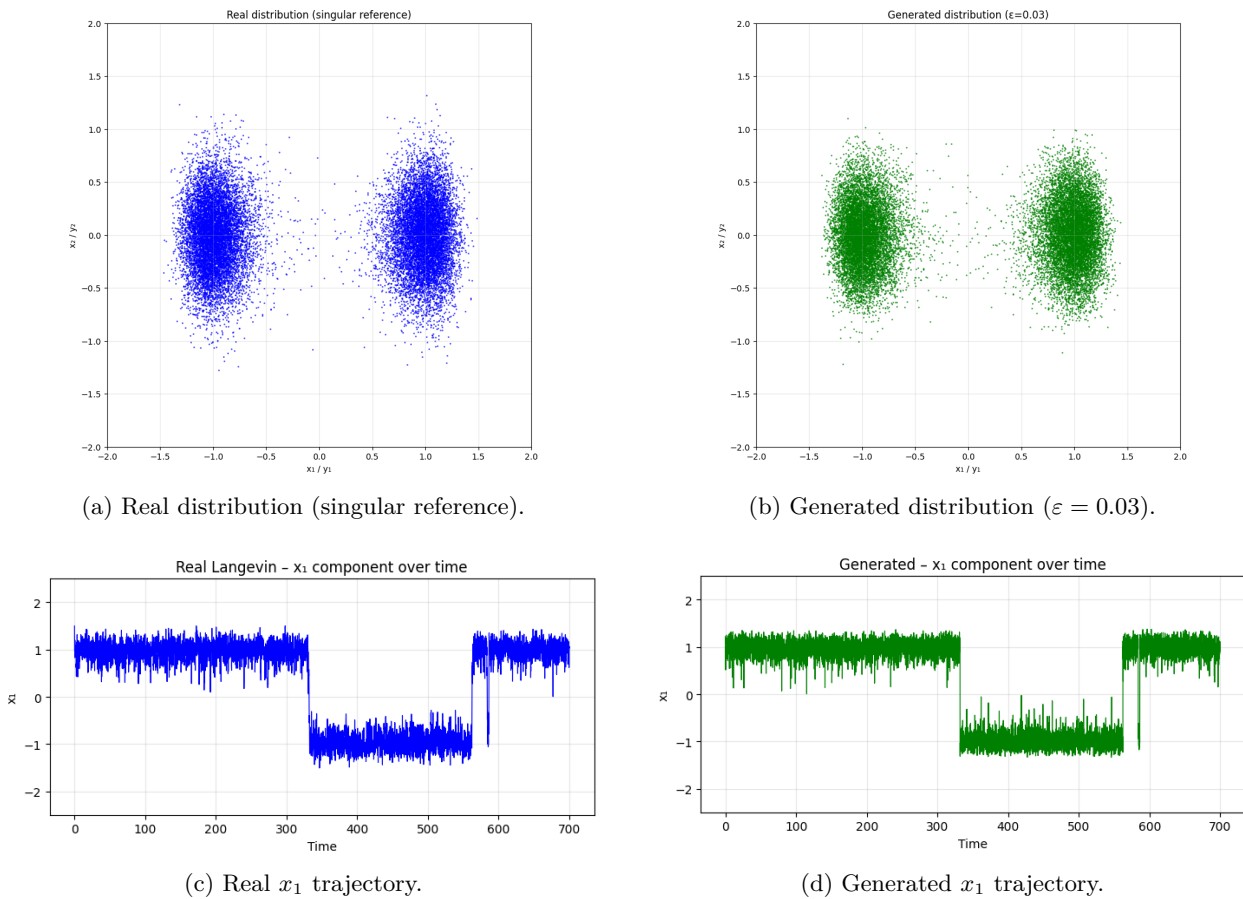

(a) Real distribution (singular reference).

(b) Generated distribution ($\varepsilon = 0.03$).

(c) Real $x_1$ trajectory.

(d) Generated $x_1$ trajectory.

Figure 2: Two-particle Coulomb system: (a,b) scatter plots of particle positions; (c,d) $x_1$ trajectories.

## 6.2 Alanine Dipeptide Molecule, collective variables

Next, we apply the (original) Boltzmann generator method on realistic molecular data by studying the standard collective variables (dihedral angles) of alanine dipeptide, a benchmark system for investigating conformational dynamics Chodera et al. (2007).

We used a data set with 6 independent 10 ns molecular dynamics (MD) trajectories (60 ns, total frames 59,994) simulated in an implicit solvent at $T = 300$ K using the AMBER99SB-ILDN force field Lindorff-Larsen et al. (2010). The backbone dihedral angles, $\Phi$ and $\Psi$, which range over $[-\pi, \pi]$, serve as the slow collective variables of the system and capture the primary conformational changes. The distribution of the angles $(\Phi, \Psi)$ is obtained from the MD trajectories. Since the angles are not constrained by singular interactions, we do not consider any regularization in this example.

We trained a RealNVP flow on dihedral angle samples to learn the equilibrium densities. We used $\ell = 6$ coupling layers with 64 hidden units for each $(s_\theta, t_\theta)$ subnetwork, resulting in 2,316 trainable parameters in total. Again we used negative log likelihood as a loss.

The results are shown in Figure 3. The model successfully captured the main stable conformations of alanine dipeptide, corresponding to the well-known metastable regions of the $(\Phi, \Psi)$ free energy landscape. The generated $(\Phi, \Psi)$ samples closely follow the true Boltzmann distribution obtained from the MD simulation. The small differences seen near transition regions are expected due to sampling noise.

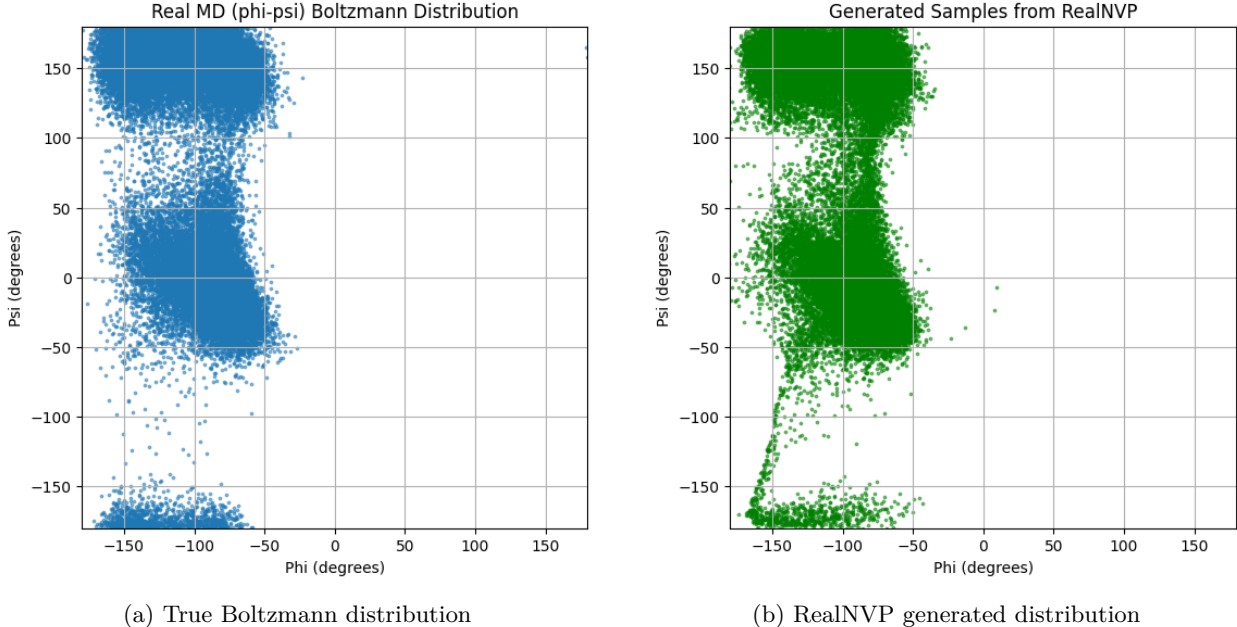

(a) True Boltzmann distribution

(b) RealNVP generated distribution

Figure 3: Comparison of true and generated $\Phi$–$\Psi$ distributions for alanine dipeptide.

To quantitatively assess the accuracy of the generated samples, as in the previous example we computed the Wasserstein-2 distance between the true and generated distributions using the Python optimal transport library Flamary et al. (2021). Following standard practice, we subsampled 1000 points from each distribution. The resulting distance of 9.21° confirms good agreement between the generated and true Boltzmann distributions, as expected from theory (Theorem 4).

### 6.3   Alanine dipeptide molecule in full 66-dimensional Cartesian coordinates

We now demonstrate the problems caused by singularities for a realistic and high-dimensional molecular system, alanine dipeptide (22 atoms, 66 Cartesian coordinates), and propose a simple ML approach to fix them. This approach bypasses the (in practice complicated) a-priori method from section 6.1 of re-generating molecular trajectories from regularized interactions. The model is trained on the original molecular trajectories but the generator is taught to avoid the collision set by adding a distance-based penalty to the loss.

The dataset is the same as in the previous section: six 10 ns MD trajectories (59,994 frames) of alanine dipeptide at 300K with the AMBER99SB-ILDN force field. The force field contains singular Lennard-Jones $r^{-12}$ repulsion between non-bonded atoms. After removing overall rotation and translation, the Cartesian coordinates are normalised to zero mean and unit variance.

Two RealNVP flows with identical architecture (12 coupling layers, 512 hidden units, 7.9M parameters) are trained for 200 epochs with the Adam optimiser. The first uses only the negative log-likelihood loss. The second adds a penalty that discourages non-bonded atom pairs from approaching too closely.

The new loss function with penalty which we propose is

$$\mathcal{L}(\theta) = -\frac{1}{N}\sum_{n=1}^{N}\Big[\log\rho_0\big(F_\theta^{-1}(z_n)\big) - \log\big|\det J_{F_\theta}\big(F_\theta^{-1}(z_n)\big)\big|\Big] \;+\; \lambda \cdot \frac{1}{|\mathcal{P}|}\sum_{(i,j)\in\mathcal{P}}\max\big(\varepsilon - r_{ij}(x),\, 0\big). \quad (17)$$

Here $\mathcal{P}$ is the set of nonbonded atom pairs, $r_{ij}$ the distance between atoms $i$ and $j$ in a generated configuration, $\varepsilon = 0.17$ nm, and $\lambda = 5.0$ controls the penalty strength.

After training, 10,000 independent samples are generated from each model. For every generated configuration we compute the minimum non-bonded inter-atomic distance $r_{\min}$ and the total Lennard-Jones potential energy $U$ of the configuration ($\sigma = 0.17$nm, $\epsilon = 1.0$kJ/mol). The results are shown in Figure 4 and Table 2.

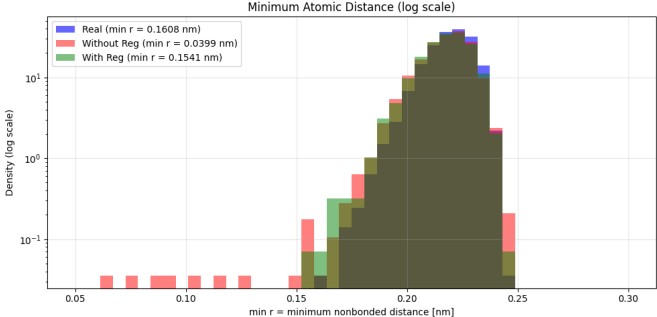

Figure 4: Distribution of $\min r$. Real data (blue), model without penalty (red), and model with penalty (green).

|  | Real data | Without penalty | With penalty |
|---|---|---|---|
| $\min r$ [nm] | 0.161 | 0.040 | 0.154 |
| $\max U$ [kJ/mol] | $-9.9$ | $1.4 \times 10^8$ | $-6.6$ |
| Mean $U$ [kJ/mol] | $-12.1$ | 30,600 | $-12.2$ |
| PCA $W_2$ (from real) | 0.0 | 0.50 | 0.76 |

Table 2: Alanine dipeptide (66D): comparison of the two trained models.

Without the penalty the flow frequently puts atoms at unphysically small distances (minimum 0.040 nm vs. the true 0.161 nm). This leads to huge Lennard-Jones energy spikes ($\max U = 1.4 \times 10^8$ kJ/mol) and a completely wrong mean energy (off by a factor $2.5 \times 10^3$). The penalty (17) shows excellent performance and removes these artefacts: the minimum distance rises to 0.154 nm, the huge energy spikes disappear, and the mean energy returns to the correct value within a relative error of 1%. During training the penalty itself quickly drops to zero, meaning that the flow learns to respect the distance constraint on its own.

To assess how well the full distribution of the real MD data is captured, we plotted the first two principal components of the configurations. Figure 4 shows that the real MD data and both generated ensembles are in good agreement. More quantitatively, the Wasserstein-2 distance of the PCA projections is very small for both flows, see Table 2. The Wasserstein-2 distance increase from 0.50 to 0.76 by the penalty is a small trade-off for the vastly improved prediction of physical quantities like energy.

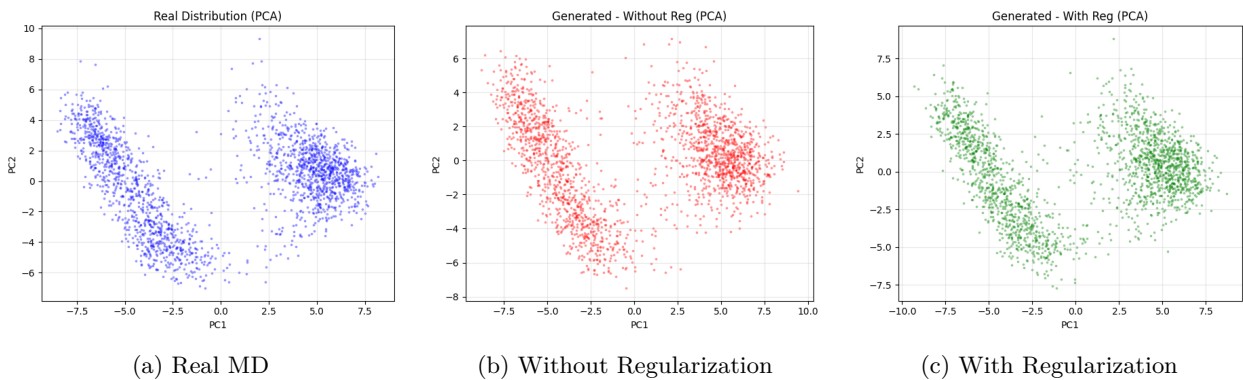

(a) Real MD       (b) Without Regularization       (c) With Regularization

Figure 4: PCA projections of the 66-dimensional alanine dipeptide configurations.

In summary, we have shown that when physical observables like energy are under consideration, the original Boltzmann generator method does not work for the standard benchmark of alanine dipeptide. This is because of the following facts: (1) energy (unlike the conformation angles studied in section 6.2) is sensitive to the density near the collision set, (2) the original Boltzmann generator method is not very accurate near the collision set. We also demonstrated how to fix this problem in practice by adding a penalty to the loss function which teaches the normalizing flow to avoid interatomic collisions.

## 7 Discussion and Conclusions

We have shown that RealNVP-based normalizing flows are capable in theory of producing correct samples from Boltzmann distributions of molecules despite their low regularity. Our theory is based on first regularizing the singular Coulomb and Lennard-Jones interatomic interactions and then using a particular transport map to the regularized distribution with nice theoretical properties (e.g., controlled Lipschitz and inverse Lipschitz constants), namely the Moser map which arises as a simplified form of optimal transport.

Our numerical simulations add further confirmation that RealNVP flows reproduce main features of Boltzmann distributions in practice as first observed in Noé et al. (2019), and quantify the error between the generated and true distributions via the Wasserstein distance.

Our simulations also indicate that current RealNVP training protocols for Boltzmann generators (which ignore the singularities) exhibit some problems in practice. They lead to large Lipschitz constants (and hence low robustness for unseen initial data), inaccurate sampling near the collision set, and vastly inaccurate estimation of observables like energy which are sensitive to the density near the collision set. The related issue of such training protocols leading to uncontrollably large Lipschitz constants for the inverse map are discussed in other contexts in Behrmann et al. (2021); Kirichenko et al. (2020).

We have made two proposals for resolving these issues, both of which show promising performance in our numerical experiments: either to regularize the singular interactions before training, or to add a penalty to the loss function which teaches the normalizing flow to avoid interatomic collisions.

An interesting question left open by our work is to optimally resolve these issues, for general Boltzmann distributions in high dimensions. Perhaps this can be achieved by a training bias towards "nice" transport maps like the Moser map without introducing too much additional computational complexity.

## Acknowledgements

This work was funded by DFG within the priority program SPP 2298 *Theoretical Foundations of Deep Learning*, project number 543965508.

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
