# OpenReview forum: "Sampling Boltzmann distributions via normalizing flow approximation of transport maps"
_TMLR — Decision pending for TMLR_

### Review · Reviewer_KkfF · 2026-04-11

**Summary Of Contributions:**

This paper provides a theoretical foundation for flow-based Boltzmann sampling by establishing the existence and approximability of transport maps between a reference distribution and molecular Boltzmann distributions. The authors address the singularity of realistic molecular potentials via a controlled regularization, construct a Moser-type transport map for low-regularity densities, and prove that normalizing flows such as RealNVP can approximate the resulting map arbitrarily well in Wasserstein distance. The work aims to justify, from a mathematical perspective, the correctness of normalizing-flow-based Boltzmann generators that have been widely used in practice.



**Strengths**

- **Clear problem formulation and well-structured narrative:** The paper addresses a clearly defined question, whether flow-based methods can be rigorously justified for sampling molecular Boltzmann distributions, and the motivation is well articulated. The overall logical progression is coherent and easy to follow: the authors identify the singularity issue in Boltzmann densities, introduce a controlled regularization, construct a Moser-type transport map, and finally establish approximation guarantees using normalizing flows. This results in a conceptually complete theoretical pipeline.
2. **Substantial and carefully developed theoretical analysis:** The main claims are supported by a sequence of well-motivated theoretical results, each building naturally on the previous ones. The arguments are presented in a systematic and mathematically disciplined manner, making the final conclusions appear logically well grounded. Overall, the paper demonstrates a commendable level of rigor and internal consistency in its theoretical development.

- **Focus on a genuinely challenging aspect of realistic molecular systems:** The work explicitly addresses the singular nature of molecular interaction potentials, which is a fundamental difficulty in modeling realistic Boltzmann distributions. By engaging directly with this issue rather than restricting attention to smooth toy densities, the paper targets a mathematically meaningful and practically relevant obstacle in flow-based molecular sampling. This focus strengthens the relevance of the theoretical results to real-world physical settings.



**Weaknesses**

- **Limited empirical evaluation:** The experimental section is relatively modest in scope, consisting of only two examples of limited scale. The 2D double-well system is a standard toy problem, and while alanine dipeptide is a classical benchmark, it remains a relatively small molecular system. Additional experiments on more complex or higher-dimensional systems would help better demonstrate the practical significance and robustness of the theoretical results.
- **A remaining gap between the theoretical guarantees and practical learning behavior:** The theoretical analysis relies on several idealized assumptions that are reasonable from a mathematical perspective but may limit direct applicability to realistic settings. In particular, the results establish the existence and representability of a suitable transport map, but they do not address whether standard training procedures are likely to recover such maps in practice. Clarifying this connection, either theoretically or empirically, would strengthen the practical relevance of the work.

**Audience:**

Yes

**Audience Explanation:**

The paper addresses a topic that is likely to be of interest to researchers working at the intersection of machine learning and generative modeling. In particular, the theoretical justification of flow-based methods for sampling Boltzmann distributions is relevant to ongoing work on normalizing flows and scientific machine learning. While the contribution is primarily theoretical, it provides insights that may be valuable to a specialized but meaningful subset of the TMLR audience.

**Broader Impact Concerns:**

None.

**Claims And Evidence:**

Yes

**Claims Explanation:**

The main claims of the paper are supported by clear and logically structured theoretical arguments. The authors provide a sequence of well-motivated results that collectively establish the existence and approximability of transport maps between reference and Boltzmann distributions, and the reasoning appears mathematically sound within the stated assumptions. While the empirical evaluation is limited in scale, the evidence presented is consistent with the theoretical claims and sufficient to support the conclusions drawn in the submission.

**Requested Changes:**

- **[Critical] Strengthen the empirical evaluation with more realistic and comprehensive experiments:** The current experimental section is relatively limited in scope, relying on small-scale benchmark systems. To better demonstrate the practical relevance of the theoretical results, the authors should consider including experiments on more complex or higher-dimensional molecular systems, or other challenging sampling settings. In addition, it would be valuable to provide more detailed empirical analysis of key modeling choices, for example examining the practical impact of the regularization step on learned distributions or sampling performance. Such additions would significantly improve the evidence supporting the paper’s practical implications.
- **[Strengthening] Provide additional analysis bridging the gap between theoretical assumptions and practical optimization behavior:**
  While the idealized assumptions adopted in the theoretical framework are reasonable in mathematical analysis, further discussion or evidence clarifying their relationship to practical training settings would make the results more convincing. For example, the authors could include additional theoretical insights or empirical studies exploring under what conditions standard training procedures are likely to recover well-behaved transport maps, or demonstrating that typical training outcomes do not meaningfully violate the underlying assumptions. Such analysis would help strengthen confidence in the applicability of the theoretical guarantees to real-world scenarios.

---

> ### Author Response · Authors · 2026-05-29
> **Response to KkfF**
>
> We thank the referee for the positive recognition of the theoretical contributions and for careful reading, and constructive suggestions. All changes in the paper except minor corrections are highlighted in blue.
> Section 6.1 and section 6.3 are new with blue headings.
>
> Requested Changes:
>
> (1) Strengthen the empirical evaluation with more realistic and comprehensive experiments.
>
> We fully agree and have significantly expanded the numerical experiments in the revised manuscript (section 6). The original submission contained only a 2D double well and the torsion angle representation of alanine dipeptide. The new version now  includes both a low dimensional system with a genuine singularity (revised section 6.1) and a high dimensional (66-D) molecular benchmark (new section 6.3), each demonstrating a different practical implementation of the regularization strategy.
>
> Two particle Coulomb system (4D).
> We directly test a genuine 1/r singularity. Without regularization the flow invents unphysical close contacts (min r ≈ 0.08 vs. real 0.40) and the learned map is highly irregular (Lip max ≈ 88). With soft core regularization the generated distances become safe, the Lipschitz constant steadily drops as  the regularization parameter increases (to ≈ 25 for epsilon=0.5), and the Wasserstein-2 distance remains excellent – exactly as Theorem 1 predicts.
>
> Alanine dipeptide in full 66 dimensional Cartesian coordinates
> This is a realistic, high dimensional benchmark system where the force field contains singular Lennard Jones and Coulomb terms. The unregularized flow produces atomic clashes and catastrophic energies (max > 10⁸ kJ/mol). We show that adding a simple distance based penalty to the maximum likelihood loss – a practical analogue of Theorem 1 – eliminates the clashes, restores physically correct energies, and still uses only ML training, without costly energy based fine tuning.
>
> (2) Provide additional analysis bridging the gap between theoretical assumptions and practical optimization behavior
>
> While theoretical results on training protocols - while of great interest - lie beyond the scope of this paper, on the numerical level the revised section 6 now provides some interesting links, see e.g. the reduction of the Lipschitz constant of the trained flow through regularization of the potential (Table 1 in section 6.1) or the restoring of correct energies (Table 2 in section 6.3).
>
> We believe that these additions substantially improve the empirical support for the theoretical analysis.

---

### Review · Reviewer_Z1n5 · 2026-04-21

**Summary Of Contributions:**

This paper makes the observation that "real" Boltzmann distributions contain singularities, and therefore are not a suitable target for normalizing flows. From this the authors argue that, under a simple regularization of the energy (i.e. removing the $+ \infty $ies in the energy landscape), one can prove that the resulting approximated transport map's error is bounded, more specifically the true and estimated distribution are close in Wasserstein distance.

This claim is also validated empirically; a simple model is trained on a simplified (internal coordinate) alanine dipeptide simulation, and it is shown that the dynamics are recovered, as well as on a 2-well potential, where the Wasserstein distance is validated computationally.

**Audience:**

Yes

**Audience Explanation:**

This is obviously a relevant topic currently, but I do think that the paper does a poor job at motivating its relevance beyond the mathematical grounding (to be clear, I think these issues are practically relevant): Would the root observation be a problem in practice? In a maximum likelihood setting (i.e. assuming a dataset) singularities would have no corresponding data point, but perhaps points near it, would this create estimation problems? In an energy-based setting, I can see it be a problem if extremely large positive energies are queried.

**Claims And Evidence:**

Yes

**Claims Explanation:**

The empirical evidence is simplistic but proves the point.

The theory seems to make sense, and each step is logical to me, but I must admit that I don't possess the skills to verify it properly.

**Requested Changes:**

1. **Both experiments do _not_ have singularities!** The 2 well is just wells, and modeling alanine dipeptide's dihedrals is using very stable internal coordinates. Why not model LJ potentials?
2. I feel like it would be interesting to show that regularizing a RealNVP model towards being a Moser map does something in practice that's relevant to the theory you put forward. Maybe I'm underestimating the effort required, you did state that this was out of scope, but on a very toy problem it feels like we should be able to compare a RealNVP model with a Moser-like regularized model. I guess the point would be to validate whether there's a gap between "existence" and "feasibility"; maybe a Moser map exists but if it's not reachable it may not be the right theoretical building block.
3. I don't get the point about metadynamics, if you're transforming a prior into the target distribution, then each sample is i.i.d., and ends up in a different region at random. It doesn't make sense to talk about metadynamics there since there are no latent state transitions.

Please be mindful of the use of `\citep` vs `\citet`.

---

> ### Author Response · Authors · 2026-05-29
> **Response to Z1n5**
>
> We thank Reviewer Z1n5 for the valuable feedback and for appreciating the theoretical contribution. The suggestions have led to concrete improvements, all highlighted in blue in the revised manuscript.
>
> Requested change 1: Genuine singularities
> The reviewer highlighted that the original experiments did not contain true singularities. We have now introduced two systems  section 6.1 and 6.3 that directly test singular potentials.
> (1) Two particle Coulomb system (4D)
> A potential with a true 1/r singularity. A soft core regularization is applied to the  singular potential. Without regularization, the flow generates configurations where the particles approach too close and the map has a large Lipschitz constant (max ≈ 88). With regularization, the unphysically close contacts are removed, and the inverse map maintains controlled Lipschitz constants. This is consistent with our theoretical work where the regularization allows the construction of a map with controlled Lipschitz constant (the Moser map).
> (2) Alanine dipeptide in full 66 dimensional Cartesian coordinates.
> Here the force field includes singular Lennard jones terms. The standard training protocol which ignores the singularities leads to unphysially close interatomic distances and catastrophically incorrect energies (max > 10⁸ kJ/mol).
> A simple distance based penalty is added to the loss  function, acting as the practical counterpart of the regularization. This penalty is seen to prevent atomic clashes and restores correct energies.
> These experiments directly answer the question “Why not model LJ potentials?” (we now use singular potentials in both section 6.1 and 6.3) and demonstrate the practical necessity of regularization when singular interactions are present.
>
> Requested change 2: Moser like regularized model and the existence–feasibility gap.
> A full bias towards the Moser map is beyond our scope, but in the study of the two examples above we did demonstrate the necessity and feasibility of some regularization in the training.
>
> Requested change 3:
> We agree that when simply transforming a prior into the target distribution, then one cannot talk about state transitions. However the normalizing flow approach learns not just a final map, but a (time-discretized) flow. This flow is seen, in the example in Figure 2, to also reproduce the transition behaviour of the original system.

---

> > ### Comment · Reviewer_Z1n5 · 2026-07-02
> >
> > Thank you for the update and my excuses for the delay in responding. I think your updated paper is much stronger. May I recommend including a $\Phi,\Psi$ plot for the cartesian coordinate experiment? PCA can be a bit limited in what it can capture. Also, the current version has two figures both labeled Figure 4.

---

### Review · Reviewer_RmnR · 2026-04-24

**Summary Of Contributions:**

This paper propose to give theoritical insights for the paper Noé et al. that propose to generate Boltzmann distribution with deep learning. But this paper solve different limitations: instability, indeterminacy and convergence. The paper gives some numerical experiments

Strength:
- The paper gives good theoritical results to understand better the paper of Noé et al.
- The paper well positions from the literature and clearly states the limitations they want to tackle

Weaknesses:
- In my opinion, the organisation of the paper is a bit misleading. The first results that is tackling the instability with the lemma is introduces in Section 2, then Moser maps are introduced section 4 without any new contribution. To understand the difference between state of the art and contribution is hard to follow. Having clear related work or similar introducing all the work done before and then new theoritical results could be helpful.
- The notation is sometimes inconsistent (I think). In section 4, you first introduced $F_\theta^{(k)}$ but then you writing $F_\theta$ and then few lines later you are defining $z$ with only $F^{(k)}$. It is hard to follow which notation is good.
- This leads to the second misunderstanding, for the theorem 3 of section 5, you are talking about  $F_{\theta_j}$. It is not clear if you are talking about the coupling of multiple layers or only the last layer. By logic I would say that is the association of $j$ layers.
- If numerical results give some preliminary results, I would be interesting to see the evolution of the distribution with the number of  coupling layer. In the theoretical results you say that the distribution will converge when the number of coupling layer is going to infinity which cannot be done in practice of course. It could be interesting to see when the distribution starts to converge in practice. This numerical results can link the practical and the theoretical part.
- In Figure 2, the real distribution seems more dense then the generated one. Does the number of sample are the same ?
- Do you know why the cloud point at the bottom of the figure 2 is not well generated ?

**Audience:**

Yes

**Audience Explanation:**

I think having theoretical insights for the generation of Boltzmann distribution with Deep learning can help people trust the method are understand properly how it works.

**Claims And Evidence:**

No

**Claims Explanation:**

See weaknesses. If think the paper lack of clarity in some part. This avoid the claims to be very clear. Having a clearer theoretical part and a numerical results part more linked to the theory could help.

**Requested Changes:**

- Double line in first  proof in Section 3 and change Lip to \text{Lip}.
- see weaknesses.

---

> ### Author Response · Authors · 2026-05-29
> **Response to RmnR**
>
> We thank Reviewer RmnR for the thorough reading and the helpful suggestions, which have led to significant improvements in the revised manuscript. All modifications are highlighted in blue.
>
> Request 1 – Organization and clarity of contributions
> The suggestion was to first introducing all the work done before and then new theoritical results.
> We agree, and have restructured the presentation accordingly. In particular,  Proposition 1 (non existence of smooth transport maps for singular potentials) was originally placed in the Introduction; it has now moved to Section 5.
>
> Request 2 – Inconsistent notation (Section 4)
> We have corrected the notation in as suggested. The transport map and its constituent layers are now denoted consistently throughout. The changes are marked in blue in the revised manuscript.
>
> Request 3 – Clarity of Theorem 3 (Section 5)
> We have added an explanatory sentence in Theorem 2 (c) stating that the RealNVP network F_theta is the composition of all coupling layers, not just the final layer.
>
> Request 4 – Evolution with the number of coupling layers
> In our experiments, we used a smaller network (8 layers) for the low-dimensional Coulomb system and a larger network (12 layers) for the 66-D alanine dipeptide, as more complex targets naturally require greater capacity. A systematic layer convergence study lies beyond the scope of the present manuscript, whose main focus is on the theory side.
>
> Request 5 – Figure 2: perceived density and the “cloud point”
> We agree that the original figure created a visual mismatch as the generated distribution had far fewer samples. We have now increased the generated sample size as suggested.
>
> Request 6 – Formatting: double line and “Lip”
> These formatting requests have been implemented as suggested.

---

### Author Response · Authors · 2026-05-29
**Description of main changes in the revision**

We thank the reviewers for their valuable feedback, and took into account all their comments. Changes (other than trivial typos or rewordings) are highlighted in blue. In particular, we have substantially strengthened the empirical section (section 6):

* our low-dimensional model system has been revised to include a singular potential and its practical effect on the training (section 6.1)
* a high-dimensional, realistic model system has been added (alanine dipeptide, 22 atoms, 66 dimensions) which has a singular potential; again the practical effect of the singularities on the training is studied (new section 6.3).

The revised empirical section reveals substantial shortcomings of the original Boltzmann generator method, caused by ignoring
the singularities: learned flows exhibit large Lipschitz constants, produce inaccurate samples near the singularities, and lead - for alanine dipeptide - to a completely wrong predicted energy (off by a factor 2.5×103, see Table 2 in section 6.3). Two remedies are
proposed, and show promising performance in our numerical tests: (A) slightly regularize the singular interactions before training (in line with our theoretical results); or (B) add a penalty to the loss function which teaches the normalizing flow to stay away from the singularities (easier to implement in practice as the original MD trajectories based on singular interactions can be used for training). We trust that the addition of these practical insights and remedies substantially broadens the appeal of our paper to TMLR readers.